# Zinc in Prostate Health and Disease: A Mini Review

**DOI:** 10.3390/biomedicines10123206

**Published:** 2022-12-10

**Authors:** Nishi Karunasinghe

**Affiliations:** Auckland Cancer Society Research Centre, Faculty of Medical and Health Sciences, University of Auckland, Private Bag 92019, Auckland 1142, New Zealand; n.karunasinghe@auckland.ac.nz

**Keywords:** Zinc (Zn), prostate, citrate, Zn transporters, metallothionines, matrix metalloproteinases, Zn fingers, Zn signaling, trace elements, transgender women

## Abstract

**Introduction**-With the high global prevalence of prostate cancer and associated mortalities, it is important to enhance current clinical practices for better prostate cancer outcomes. The current review is towards understanding the value of Zn towards this mission. **Method**-General information on Zn in biology and multiple aspects of Zn involvement in prostate health and disease were referred to in PubMed. **Results**-The most influential feature of Zn towards prostate health is its ability to retain sufficient citrate levels for a healthy prostate. Zn deficiencies were recorded in serum, hair, and prostate tissue of men with prostate cancer compared to non-cancer controls. Zn gut absorption, albumin binding, and storage compete with various factors. There are multiple associations of Zn cellular influx and efflux transporters, Zn finger proteins, matrix metalloproteinases, and Zn signaling with prostate cancer outcomes. Such Zn marker variations associated with prostate cancer recorded from biological matrices may improve algorithms for prostate cancer screening, prognosis, and management when coupled with standard clinical practices. **Discussion**-The influence of Zn in prostatic health and disease is multidimensional, therefore more personalized Zn requirements may be beneficial. Several opportunities exist to utilize and improve understanding of Zn associations with prostate health and disease.

## 1. Introduction

Global cancer statistics show that prostate cancer is the second most common cancer type and the fifth leading cause of cancer mortality in men [1]. Greater predictive accuracy of prostate cancer diagnoses, prognoses, and treatment, alongside standard clinical frameworks may therefore better support prostate cancer outcomes. Multiple non-modifiable and modifiable risk factors of prostate cancer have been reported with smoking and obesity consistently being associated, especially with advanced disease [2,3]. Wilson and Mucci 2019 reviewed mostly prospective cohort studies to understand the influence of various dietary and lifestyle factors on prostate cancer risk, its aggressiveness, and survival [4]. These authors have noted that apart from dairy intake associated with increased risk, and fish intake and lycopene/tomato intake associated with lower risk, other dietary factor influences are inconclusive due to their reported heterogeneity in risk outcomes.

Excessive dietary intakes as in the Westernized diets however produce imbalanced levels of reactive oxygen species (ROS), reactive nitrogen species (RNS), and lipid oxidation (LOX) moieties causing damage to biomolecules, especially when insufficiently counteracted through the action of antioxidant systems [5]. Alongside vitamin C, vitamin E, glutathione (GSH), melatonin, alpha lipoic acid, and carotenoids, trace elements such as zinc (Zn), copper (Cu), and selenium (Se) are considered among non-enzymatic antioxidants in living organisms [6]. Unless the enzymatic and non-enzymatic antioxidant fractions can counteract the excess ROS, RNS, and LOX products, damage to the biomolecules becomes inevitable, leading to diseases more prevalent in Westernized societies, including cancer types such as prostate cancer. The influence of excess dietary intake and the consequences of oxidized food intake on prostate cancer are reviewed in Estevez et al., 2017 [6]. The ability of trace elements to counteract biomolecule damage is only part of their functions as many other functions associated with trace elements are known to date. The following review wishes to collate scientific information on how Zn is associated particularly with prostate health and prostate disease functions and to understand how this knowledge may be applied to benefit prostate cancer prevention, diagnosis, prognosis, and management.

## 2. Method

General information on Zn in biology and multiple aspects of prostate health and disease associations with Zn were referred to in PubMed articles and references therein. For some comparisons, the Zn association in breast cancer and work involving prostate cell lines and animal models were also reviewed. The selection of articles had no publication time limitation and therefore historical articles to the most recent up to November 2022 were reviewed. Search features included serum Zn levels and prostate cancer; tissue Zn levels, distribution, and prostate cancer; mitochondrial *aconitase* (*mACO2*) and prostate; citrate and prostate; Zn absorption from gut and circulation in the blood and competition for albumin binding sites; Zn cellular and sub-cellular transporters, Zn influx transporters, and Zn efflux transporters; Zn storage in metallothionines and competition for binding sites; Zn regulation of matrix metalloproteinases and effects on prostate cancer; Zn finger proteins and effects on prostate; Zn signaling and prostate health; involvement of Zn in prostate associated proteins (prostate-specific antigen (PSA), prostate secretory protein of 94 amino acids (PSP94) also known as β-microseminoprotein, the unique human-specific transmembrane metalloprotease prostate-specific membrane antigen (PSMA) and prostatic acid phosphatase); single nucleotide polymorphisms (SNPs) of genes encoding Zn associated proteins or Zn interacting trace element associated proteins; Zn supplementation effects; interaction with other trace minerals, demographic, lifestyle and comorbidity effects; transgender women and prostate cancer; and Zn dependent technologies associated with prostate cancer.

## 3. Results

### 3.1. Overview of Zn Proteins and Circumstances for Prostate Malignant Transformation

The human genome encodes over 2800–3000 Zn proteins, representing approximately 10% of the human proteome [7,8]. Zn is considered the most versatile in metalloenzymes found in living organisms and represents over 300 enzymes with more than 50 different types known to require Zn for their function [9]. According to these authors, all six classes of enzymes established by the International Union of Biochemistry namely oxidoreductases, transferases, hydrolases, lyases, isomerases, and ligases are represented by Zn. 

Zn deficiency-related outcomes were first recorded in hypogonadism and dwarfism in males in 1963 [10]. Tissues with high cell turnover such as the gonadal tissue are more sensitive to Zn deficiencies [9]. The authors also provide a review of unique Zn binding sites in enzymes associated with catalytic, co-catalytic, and structural functions as well as the major Zn protein groups such as matrix metalloproteinases, metallothionines (which may also contain cadmium (Cd), Cu, Fe or mercury (Hg)), and the gene regulatory proteins. Metallothionines in particular act like a type of reservoir for Zn^2+^ ions and are capable of releasing them under oxidative stress conditions [11]. 

Metabolic transformation to prostate malignancy is associated with normal Zn-accumulating epithelial cells altering to malignant cells that have lost the ability to accumulate Zn [12,13,14,15]. This transformation is considered due to normal cell Zn levels being cytotoxic to the malignant cells and is prevented from accumulation [15]. The well-known redox-sensitive tumor suppressor p53 protein is a Zn-binding protein in which Zn is essential to maintain the protein folding feature, and a large proportion of tumorigenic p53 mutations cause this protein to lose Zn [16,17].

### 3.2. Zn Absorption from Gut and Blood Circulation

The recommended dietary allowance (RDA) of Zn is estimated as 11 mg/d for men over 19 y [18]. This level has been set up carefully considering multiple variable factors including varying excretion rates through feces and absorption rates, while urinary excretion and losses from sweat and semen have been considered constant [19]. To keep up with Zn homeostasis and to replenish endogenous Zn losses, absorption of dietary Zn by the enterocytes of the gut lumen mainly happens in the duodenum and jejunum of the human small intestines [20]. The authors also provide a general overview of this process including the inhibitory impact of dietary calcium (Ca), cadmium (Cd), iron (Fe) and tin (Sn), phytate, and heat-derived zinc-binding ligands, such as Maillard browning products; Zn absorption being supported by citrate and fermented and germinated products; intestinal Zn bioavailability; Zn uptake at the apical membrane of the intestinal mucosa and absorption through the enterocytes and transfer to the blood albumin at the portal vein; types of Zn pools absorbed and the requirement of strict control of free Zn^2+^; Zn excretion; the need of Zn distribution to different tissues to fulfill their requirements; and to prevent deficiencies related diseases; and recent advancement of Zn absorption modeling at the gut barrier. Zn absorption at the gut may also be impacted by increased transient time as in the case of malabsorption syndromes [19], while low-fiber diets may also produce such absorption impacts. Serum carries 14.9 ± 2.4 µM of Zn in men mostly bound to serum albumin (12.5 + 2.1 µM of Zn) [21,22], while the rest are bound to other proteins such as 2-macroglobulin, haptoglobin, ceruloplasmin, immunoglobulins (IgG, IgM, and IgA), complement C4, prealbumin, C-reactive protein, and fibrinogen [23,24,25,26,27,28]. According to Costello and Franklin 2020, the source of Zn that is available to be transported into the cells is what is available in the interstitial fluid with a concentration of 2-4 µM [15].

The crystal structure of serum albumin shows a major Zn^2+^ binding site involving three protein ligands from the sidechains of His67, His247, and Asp249 and a water molecule, besides two more secondary and 15 more tertiary binding sites with decreasing binding affinities [29]. The major site is a multi-metal binding site and can also bind other metal ions such as Cu^2+^, Ni^2+^, Co^2+^, and Cd^2+^ while having a higher affinity towards Zn^2+^ [30,31]. Various binding affinities of these metal ions are also described [29], indicating competition for Zn^2+^ binding at these sites. Zn binding to albumin also competes with free fatty acids [32], and free fatty acid binding changes the structure of serum albumin and reduces its affinity for Zn^2+^ [33]. Meanwhile, prostate cancer patients report higher levels of free fatty acids in the serum compared to men without prostate cancer, and in vitro studies with PC3 (cell line derived from human prostate cancer bone metastasis) and 22RV1 (prostate carcinoma epithelial cell line derived from a xenograft) cells have shown that free fatty acids promote the proliferation, migration, and invasion of these cells [34]. Whether the higher levels of free fatty acids in the serum impact prostate outcomes by impacting Zn carrier capability is however not yet confirmed. 

Level of the Ischemia-modified albumin (IMA) is higher in men with prostate cancer compared to healthy controls, and the levels are higher in men diagnosed with Gleason grade ≥7 cancers compared to those with Gleason grade < 7 cancers; and cases with Stage 2-4 disease compared to those recording Stage 1 prostate cancer [35,36]. Meanwhile, a decrease in serum Zn levels associated with increased IMA levels has been so far reported in psoriasis patients [37]. Patients with hypertension record an increase in IMA levels, Cu, Fe, and superoxide dismutase activity, malonaldehyde levels, and a corresponding decrease in Zn levels [38]. Although IMA levels are implicated in many cancers including prostate cancer [39], the interaction of serum IMA and Zn levels in cancer is not yet recorded.

### 3.3. Zn Tissue Accumulation and Functions

The hippocampus and neocortex region of the brain, islets of Langerhans of the pancreas, and the prostate gland, are high Zn accumulating organs in the body, and their Zn stores are used for secretory pathway requirements and the latter, especially for citrate production [40,41]. A meta-analysis indicates that Zn levels in breast cancer tissue are higher than that of the healthy tissue while serum levels are lower in breast cancer patients compared to the controls [42]. According to a meta-analysis, serum Zn levels are also lower in prostate cancer patients compared to men with normal prostate or in those with benign prostate hyperplasia (BPH) [43], although levels diminish in prostate cancer tissue compared to healthy prostate tissue [44]. Higher serum Zn concentrations in BPH patients compared to normal controls are also recorded [43]. Changes taking place in Zn pools from normal to malignant prostate have also been reviewed [45]. 

Prostate glandular epithelial cells carry an exceptionally higher level of citrate compared to any other soft tissue [46]. The importance of Zn in the prostate is considered as a requirement to inhibit m*ACO2* activity towards truncating the Krebs cycle at the citrate oxidation step to assure required citrate accumulation in prostate epithelial cells [47,48]. Prostate glandular epithelial cells are thus capable of survival with an energy-efficient status with healthy citrate levels, although in a lower Zn environment the full activity of m*ACO2* takes place, and this stability disruption leads to the completion of the Krebs cycle with the production of excess energy benefitted by the potential malignant cells for their survival [49]. Xue et al., 2019 have shown that the m*ACO2* levels were higher in prostate cancer tissue compared to the adjacent non-cancer tissue and correlate positively with prostate cancer malignancy [48]. It is believed that m*ACO2* is sensitive to reactive oxygen species (ROS), and the function of Zn-associated inhibition of m*ACO2* occurs through free radical formation together with the action of the p53 protein [48]. 

According to Costello and Franklin 2006, the metabolically transformed citrate-oxidizing prostate cancer cells lose the ability to accumulate Zn [14]. The authors record Zn levels of 3000–4500, 400–800, and 200–400 nmol/g in the normal prostate peripheral zone, malignant prostate tissue, and other tissue, respectively, while prostatic fluid contains 8000–10,000 nmol/g. In wet weight terms, levels of Zn in the normal prostate peripheral zone, malignant peripheral prostate tissue, and other tissue, respectively are 295, 30–50, and 30 mg/kg [15]. A systematic review of Zn as a wet mass in prostates of men without prostate cancer recorded in the past century has varied between 17 to 547 mg/kg (median of means 109 mg/kg) [50]. The authors describe the possibility of measurement method variability, prostate gland content measured (epithelium, stroma, or glandular lumen), and standards used in measurement procedures to be among the reasons for this variation. Using the LNCaP (derived from a prostate cancer lymph node metastasis) and PC3 prostate cancer cell line models, it has been shown that they are still capable of rapid Zn uptake against the low concentrations available in the medium [51]. The authors further record that this uptake is stimulated by physiological concentrations of prolactin and testosterone in the medium. Meanwhile, the less aggressive LNCaP cells can accumulate more Zn than the relatively more aggressive PC3 cells, and when grown in vitro for 24 h incubated with 1 mg/mL Zn produced a 4.9-fold and 1.8-fold increase in the G2/M cell populations, respectively and a 51% and 23% decrease in the S phase cell populations, respectively compared to no Zn treated control cells [52]. The authors also recorded a lower 50% inhibition (IC 50) of 100 ng/mL for LNCaP cells and a higher IC50 of 700 ng/mL for PC3 cells. 

A higher level of Zn has been reported from the glandular epithelial cells of the prostate compared to the stroma in normal prostate tissue, while with malignancies, the levels in the glandular epithelium appears lower than that of the stroma [53]. Multiple studies record low prostate tissue Zn content in association with prostate cancer progression and severity, impairment of maintaining healthy prostate; interaction of Zn with insulin-like growth factor 1 (IGF-1), and its binding protein IGFBP-3; and association with total PSA level as reviewed by Li et al., 2020 [54]. The authors also claim that the reduced Zn level association with prostate cancer can be used as a biomarker for prostate cancer diagnosis. In situ Zn staining methods have further provided evidence that malignant cells carry decreased Zn levels compared to the normal prostate peripheral zone cells as reviewed by Costello and Franklin 2016 [45]. It has been shown that the mobile Zn accumulation within mitochondria is also affected in prostate cancer cells compared to that of healthy prostate cells [55]. 

The ability to differentiate between men with and without prostate cancer using serum levels has also been reviewed by Li et al., 2020 [54]. Although seven studies reviewed showed a decrease in serum Zn levels in prostate cancer cases compared to healthy controls, two have shown no difference. It is a possibility that there are variable serum Zn levels in these healthy controls from different geographical locations, while undiagnosed malignancies contaminating these controls may have also influenced the analyses.

Zn^2+^ binds with the seminal plasma protein Semenogelin I (Sg I) and supports initial sperm clotting which later undergoes proteolysis by PSA before fertilization of an ovum [56]. The authors show that the origin of Zn ^2+^ source in this process is also from the prostate.

### 3.4. Cellular Zn Moieties

#### 3.4.1. Zn Cellular and Sub-Cellular Transporters 

Cellular Zn level homeostasis is controlled by the Zn importer proteins (ZIP - Zn-regulated, Iron-regulated transporter-like proteins (encoded by SLC39A genes)) and the Zn exporter proteins (ZnT (encoded by SLC30A genes)) which are involved in the influx to the cytoplasm and efflux from the cytoplasm, respectively and managing Zn storage in various organelles and vesicles to supply Zn to various proteins that require Zn for their function [57,58,59,60,61]. The most recent account on Zn transporters is given in Kambe et al., 2021 [57]. 

##### Zn Influx (Import) Transporters

The first point of Zn absorption happens at the enterocytes with the support of the ZIP4 import transporter [62]. The major Zn uptake transporter proteins involved in the extraction of Zn from the circulation are ZIP-1, 2, 3 and 4 [63,64]. Albrecht et al., 2008 recorded the expression of genes coding for seven ZIP transporters (ZIP-1, ZIP-3a, ZIP-6, ZIP-7, ZIP-8, ZIP-10, and ZIP-14) in healthy prostate tissue samples, non-tumorigenic RWPE1 cell line, and prostate cancer cell lines, while the latter recording expression for the ZIP-3b as well [65]. The authors recorded that level of the ZIP10 is higher in the immortalized non-tumorigenic RWPE1 cell line compared to the healthy prostate tissue and prostate cancer cell lines. Bulldan et al., 2018 record the presence of the ZIP-9 transporter from the LNCaP cell line [66]. The ZIP-1 encoding gene expression is downregulated with a corresponding reduction in Zn level in prostate cancer tissue compared to normal prostate tissue [53,67]. It is also recorded that adjacent non-cancerous tissue to showing a gradient of Zn level depletion towards the prostate cancer lesions, indicating a possibility of lower Zn retention as a pre-malignancy event [53]. Studies with the non-tumorigenic RWPE1 and the tumorigenic RWPE2 human prostate epithelial cell lines have shown that the latter shows less intracellular Zn levels associated with a lower ZIP-1 protein expression (while the ZIP-1 encoding gene expression remains the same in both cell lines) [68]. When these RWPE2 cells were transfected with a ZIP1-Myc expressing plasmid, that resulted in an elevation of intracellular Zn concentration and corresponding growth suppression, and increased apoptosis. With the recording of the ZIP-3 protein exclusively located in the lysosomal compartment of the RWPE2 cells, the authors also note the catabolism of the ZIP-3 protein in tumorigenesis [68]. Chen et al., 2012 report that both mRNA and protein expression of the ZIP-4 were lower in prostate carcinoma tissue compared to BPH [69]. The authors also report that proliferation, migration, and invasion of the DU145 prostate cancer cells (derived from a prostate cancer brain metastasis) were reduced by transfection with the ZIP-4-containing vectors, while short hairpin RNA (shRNA) for the ZIP-4 in the prostate cancer cell line 22RV1 was associated with increased cell proliferation and invasion. Although studies indicate a general reduction of ZIP proteins with prostate cancer malignancy, a correlation between ZIP protein expression levels and prostate cancer grades is not yet recorded [70]. Functional variability between various ZIP proteins toward cancer outcomes is also a possibility. Studies with breast cancer cell lines and zebrafish embryos indicate the function of the ZIP10 and ZIP6 in epithelial-to-mesenchymal transition (EMT) [71]. The authors also demonstrate a heteromer formation between the ZIP10 and ZIP6 and consider this as facilitating EMT. Various aspects of breast cancer are associated with Zn influx transporters including breast cancer risk correlation with the ZIP6, ZIP7, and ZIP10 expression level; association of cancer cell invasion and metastasis with the ZIP6 expression; the estrogen receptor expression and disease aggressiveness and metastasis association with the ZIP6 expression; association of the ZIP-7 with high-risk disease and poor prognosis are reviewed by To et al., 2020 [70]. However, such information is not available for prostate cancer. Meanwhile, there is a crosstalk between testosterone and citrate production, and the androgen receptor-regulated expression of the Zn transporter ZIP-1, as well as the synthesis of citrate through the support of aspartate transporter the SLC1A1 in prostate epithelial cells [51,72]. The testes of the oxidative stress-induced obese mouse model, and the oxidative stress-induced mouse spermatogonia C18-4 cell line resulted in suppression of the ZIP-12 Zn transporter levels [73]. However, similar oxidative stress impacts on prostate-specific ZIP transporter levels are not yet reported. An in vitro study with the MCF-7 breast cancer cell line model has shown that high glucose levels can support survival under hypoxic stress by modulating Zn^2+^ homeostasis by decreasing the ZIP6 expression [74]. It will be of value for such studies with prostate cancer. Zn import transporters ZIP-4, ZIP-8, and ZIP-14 also import Cd to cells [75] and may impact intracellular Zn stored in metallothionines.

It is known that African American men develop prostate cancer at a significantly early age, with higher-grade adenocarcinomas at diagnosis, and higher prostate cancer mortality, when compared with age-matched European American men [76]. The gene expression level of the ZIP-1 and ZIP-2 Zn transporters was lower among the African American patients when compared with age-matched and Gleason score-matched specimens obtained from the European Americans [77]. The authors also recorded similar observations in men without prostate cancer between the African American and the European American men indicating an ethnic variability of the ZIP-1 and ZIP-2 Zn transporter encoding gene expression. 

##### Zn Efflux (Export) Transporters

Zn-export transporters recorded from various human and mouse tissue are ZNT-1, ZnT-2, ZnT-3, ZnT-4, ZnT-5, ZnT-6, ZnT-7, ZnT-8, and ZnT-10 which are reported from the cell membrane and membranes of intracellular organelles and they function towards the transfer of cytoplasmic Zn to the extracellular matrix or lumens of cell organelles [78,79,80,81,82,83,84,85,86]. Expression of mRNA for the export transporters ZnT-1, ZnT-2, ZnT-4, ZnT-5, ZnT-6, and ZnT7 are recorded from human prostate cancer tissue, mouse prostate tissue, non-tumorigenic RWPE1 cell line, and prostate cancer cell lines [65,87]. 

Regulated by cellular Zn levels, the ZnT-1 functions towards intracellular Zn balancing by export to the extracellular spaces and avoids Zn accumulation leading to toxicity [88,89,90]. Based on the Oncomine database records, the expression of Zn-exporter genes SLC30A1, SLC30A9, and SLC30A10 are upregulated, and SLC30A5 and SLC30A6 are downregulated in prostate cancer compared to BPH [90]. The authors also report that the ZnT-1 (the only member of the SLC30 family that exports the cytoplasmic Zn^2+^ across the cell membrane to the extracellular space) showed no correlation with tumor stage, while only those with the Gleason grade 3+4 showed an association with this Zn transporter. The authors further show an ethnic difference in upregulation of SLC30A1 expression in prostate cancer tissue compared to BPH; where this is found only in European Americans, while African Americans show no difference, and the expression level getting downregulated in all prostate cancer cell lines irrespective of their racial origins. Hasumi et al., 2003 utilized prostate tissue sourced from Japan, where the malignant cells recorded a downregulation of ZnT-1 expression compared to BPH [91]. Therefore, ZnT-1 level in association with prostate cancer tissue seems to vary with ethnicity. A reduced ZnT-1 expression is associated with androgen-independent LNCaP-AIDL prostate cancer subline (generated by culturing prostate cancer LNCaP cells in a hormone-deprived medium [92]) compared to androgen-responsive LNCaP cells [93]. Expression of genes encoding ZnT-1 was upregulated by Zn supplementation in both LNCaP and PC3 prostate cancer cell lines and the corresponding protein expression upregulation was confirmed in PC3 cells [91]. 

A rodent study has shown comparatively higher expression of ZnT-2 in the dorsolateral prostate compared to the ventral lobe, and Zn deficiency mainly affects the former over the latter [94]. The authors also record that out of the Zn-export transporters ZnT-1-4, and import transporters ZIP-1-4 studied, it was only ZnT-2 that showed a correlation with Zn level in the dorsolateral lobe. A trend towards higher ZnT-2 levels is reported from prostate cancer tissue of African American men compared to European American men and for stage III prostate cancer compared to other stages [90]. However, due to the lower number of samples expressing ZnT-2 in the tissues studied, the authors consider this as not conclusive. A study with LNCaP cells and its androgen-independent subline AIDL cells show higher expression levels of ZnT-3 levels [93]. Zn transporters such as ZnT-2 and ZnT-4 recorded from lysosomes [85,95] could be showing a major Zn detoxification pathway. According to Kikic et al., 2014, Zn^2+^ accumulation in lysosomes to prevent toxicity levels in the cytoplasm will be tolerated only up to a threshold and thereafter is removed to the extracellular matrix through lysosomal acidification and exocytosis, and prevention of this process leads to cell death [95]. Molecular profiling carried out in 74 tissue samples extracted from radical prostatectomy, and 164 normal tissue samples representing 40 different body tissues, including four normal prostate samples has shown gene clusters specific for prostate cancer and normal prostate. This study records a threefold higher level of ZnT-4 gene expression levels in normal prostate compared to other normal tissue, and a fivefold higher level in prostate cancer compared to other types of male tissue analyzed [86]. Additionally, the authors also record that ZnT-4 protein expression in LNCaP cells, but not in PC3 cells, and the expression is moderate to high in most of the normal and BPH tissue while prostate cancer tissue recorded a higher number of samples (75%) with weak staining. However, they saw no correlation between ZnT-4 levels and prostate cancer outcomes such as the Gleason score, seminal vesicle invasion, positive surgical margins, and pre-operative PSA.

ZnT-5 is highly expressed in the human prostate [96], while ZnT-6 is reported in the mouse prostate [87]. Osteoblast maturation to osteocyte is affected in mice deficient in ZnT-5 resulting in multiple bone impairments [96]. This could imply that ZnT-5 deficiency may potentially have serious compound consequences in bone health in men treated with androgen deprivation therapies (ADTs or hormone therapies) used to treat advanced prostate cancer and are known to affect bone stability [97,98]. As discussed in Singh et al., 2016 and references therein, although ZnT-6 is not capable of Zn transport on its own, this dimerizes with ZnT-5 for the transporter function and its translocation from the perinuclear region to cell periphery is noted under high Zn levels [90]. However, no studies have been reported on ZnT-6 association with prostate cancer. A transgenic adenocarcinoma of the mouse prostate (TRAMP) model with a Znt7-null genetic mutation has led to higher frequencies of low-grade prostatic intraepithelial neoplasia (PIN) and high-grade PIN at 6–8 weeks, higher frequencies of prostate tumors and their metastasis at 16 weeks, and worsening of prostate tumor frequencies at 28 weeks compared to the TRAMP mice [99].

#### 3.4.2. Metallothionines

Metallothionines (MTs) are the major low molecular weight intracellular metalloproteins that are ubiquitously distributed, equipped with structures unusually high in cysteine content with thiol (SH) groups providing seven binding sites for Zn and Cu ions for storage, distribution, and release, and for Cd ions for the sequestration and to be finally accumulated mainly in the proximal tubules of the kidneys and to a lesser degree in the liver [75,100]. Thomas et al., 1992 carried out a Zn depletion and supplementation study with humans and confirmed the responsiveness especially of erythrocyte MT to dietary zinc intake [101]. Accumulation of Cd occurs mainly bound in MT (Cd-MT) and Cd binding to MT is stronger than that of Zn and therefore can easily occupy most of the seven metal binding sites in the case of environmental exposure [75]. These authors, also provide a comprehensive review and commentary on Cd-MT toxicology including, the inverse association between Zn-bound MT and Cd-MT, the importance of the Cd/Zn ratio in MT for toxicity manifestation in the kidneys, binding of other metals/metalloids such as Se and bismuth (Bi) to MT in vivo, the four forms of MT 1–4 that have been identified, routes of Cd body entry, long-term Cd exposure favoring Cd storage in MT, biological half-life in humans (10–30 years), minor excretion through urine, ROS effects associated with Cd toxicity, toxicity and toxicokinetic modeling of Cd. Meanwhile, various impacts of Cd on prostate cancer outcomes are known including associations with overall and PC-specific mortality risk and aggressiveness [102,103]. 

A meta-analysis on histochemical staining of MT on cancer tissue samples vs. normal tissue has shown differential levels of staining in certain cancers, but not on prostate and breast cancer, although strong associations were observed with tumor grades of prostate and breast cancer [104]. 

#### 3.4.3. Matrix Metalloproteinases 

Matrix metalloproteinases (MMPs) are a large family of multidomain Ca-dependent Zn-containing endopeptidases well known for their protease activity on the degradation of extracellular matrix (ECM), although other functions of MMPs are also known [105,106,107,108]. A total of 24 genes encoding for human MMPs are known [109]. The active site of the MMP is inactive when occupied by the MMP propeptide domain coordinated with Zn^2+^ binding at the cysteine (99th) sulfhydryl residue [110,111]. As a cofactor for MMPs, Zn is important for its expression and activity, with higher Zn levels reducing, and deficiencies increasing its activity [112,113,114,115]. MMP involvement in cancer is not only due to tumor metastasis with physical degradation of ECM, but also due to factors such as ECM degradation exposed integrin family of cell surface receptors promoting cell proliferation, and exposed signaling components and growth factors supporting tumor development [116]. 

Binder and Ward 2021 give a systematic-like review of MMPs and prostate cancer progression [117]. The authors give a comprehensive account of multiple MMPs from three superfamilies (Matrixin, Astacin, and Adamalysin) that either promote or suppress prostate cancer proliferation, invasion, and metastasis as well as those associated with survival from prostate cancer and chemoresistance. The authors also note that the most extensively studied are the soluble MMPs, mostly with protumorigenic features including MMP-2, MMP-7, and MMP-9. 

#### 3.4.4. Zn Finger Proteins

A review by Neuhaus 2022 gives an account of the discovery of Zn finger (ZF) proteins, their varied structures, and DNA, RNA, and protein binding [118]. The author gives a loose description of ZF proteins as those with domains whose structures and functions have been stabilized by one or more tetrahedrally co-ordinated Zn^2+^ ions bound to different combinations of cysteine (C) and histidine (H) residues (typically two Cs and two Hs) without getting involved in its enzyme chemistry. Among such proteins are a large number of Zn finger-containing transcription factors [119]. Mackeh et al., 2018 give an account of the evolutionarily conserved transcription factors such as nuclear hormone receptors which get further subjected to more orderly regulation in higher life forms through transcriptional regulators/cofactors presented by multiple Zn fingers [119]. Among such are the Zn finger-carrying specificity protein (SP) family of which the most studied the SP1 regulates the transcriptional activity of reproductive system hormone receptors including the androgen- receptor (AR) by binding to the GC-rich AR promoter [119,120]. AR receptor is highly expressed in both primary and metastatic prostate cancer and patients with such higher AR levels are destined for poor prognosis with early disease recurrence after primary treatment, resistance to androgen deprivation therapy as well as disease progression [121]. Meanwhile, AR itself is a Zn finger protein class with Zn binding to four cysteine residues [122]. For more details on prostate cancer association with AR and SP1 Zn finger domains as well as the promyelocytic leukemia zinc finger (PLZF) carrying multiple adjacent-Zn finger domains, the reader is directed to Li et al., 2020 [54]. 

The Zn finger C3H1 domain-containing protein (ZFC3H1) is considered associated with the retention and degradation of polyadenylated RNA and preventing their cytoplasmic transport for subsequent translation [123]. Using a bioinformatics approach on The Cancer Genome Atlas Database, Huang et al., 2021 report that the ZFC3H1 domain-containing protein expression is lower in prostate adenocarcinoma compared to adjacent non-cancerous tissue, although men recording higher expression levels were associated with lower survival [123] indicating a ‘U’ shaped cancer outcomes with ZFC3H1 levels. The authors also report that the knock-down of ZFC3H1 in prostate cancer cell lines 22RV1, and DU145 lead to the inhibition of cell migration and invasion, and reduces cell viability while increasing apoptosis. Oxidation of Zn finger cysteine residues can impair the Zn finger structures [124]. Impaired Zn finger binding forces associated with nitrogen oxide and DNA mutagenesis are also known [125,126].

Zn finger protein ZNF185 is a LIM domain gene [127] the expression of which is downregulated in several epithelial cancers including prostate cancer [128]. According to Smirnov et al., 2018, ZNF185 is an actin cytoskeleton-targeting protein that is induced following DNA damage and is a p53 target gene [128]. Transcriptional silencing due to the 5’CpG methylation of this gene is reported to be associated with prostate cancer progression and this methylation was recorded from the tested prostate cancer cell lines, in all metastatic and 44% of the localized prostate cancer tissues [129]. The authors also report restoration of the ZNF185 expression levels with DNA methylation inhibitors.

Considering multiple aspects of Zn finger protein signaling domains, they carry the potential to be developed as prognostic markers as well as therapeutic targets in prostate cancer [54,130].

### 3.5. Zn Signaling

Zn signaling with the release of Zn^2+^ from proteins such as metallothionein or cytosolic import through ZIP transporters is known for multiple intracellular signaling pathways [131,132,133,134,135,136]. Li et al., 2020 provide a review of advances in Zn signaling and modulation of persistent cell proliferation, resistance to apoptosis, induced angiogenesis, and invasion and metastasis associated with prostate cancer [54]. A graphical summary from Li et al., 2020 [54] review is presented here with the publisher’s permission (Figure 1). Li et al. and references therein describe the involvement of Zn in regulating cell proliferation and growth through the activation of the extracellular signal-regulated kinase 1/2 (ERK1/2) phosphorylation through the Vaccinia H1-related phosphatase (VHR)/Zeta chain-associated protein-70 (AP-70-associated) pathway, regulating phosphorylation of the tumor suppressor PTEN and associated genes, decreasing AR expression, expression of the proliferative gene IGF-1, increasing p21 expression; regulating apoptotic pathways including the release of cytochrome C from the mitochondria into the cytosol, mitochondrial apoptosis, activation of a cascade of caspases, decrease expression of the anti-apoptotic or pro-survival genes, activated p21 expression; regulating cancer cell invasion and metastasis through pathways including intercellular adhesion molecule(ICAM-1) and other angiogenic and metastatic factors, vascular endothelial growth factor (VEGF), IL-8 and MMP-9, reducing collagen type IV degrading aminopeptidase N (AP-N), and controlling hypoxia-inducible factor-1 (HIF-1), inhibitory activity of nuclear factor kappa B (NF ĸ-B) and activating immune cells within the tumor microenvironment. Readers can refer to Li et al. [54] for more details on Zn signaling events of relevance to prostate cancer. 

Using fresh prostate cancer tissue Zhang et al., 2022 have shown that advancing prostate cancer is associated with mitochondrial complex II-dependent ROS production and impaired mitochondrial complex I oxidation, which originates from insufficient suppression of mitochondrial aconitase due to Zn deficiency [137]. Nuclear factor erythroid 2-related factor 2 (NRF2) is the master regulator for oxidative stress management, and the activation of NRF2-antioxidant responsive element pathway enhances Zn efflux from cell organelles including that of mitochondria [138]. Therefore, while Zn deficiency is associated with excess ROS production in mitochondria of the prostate epithelial cells, ROS production also leads towards impairing further Zn access to the mitochondria to truncate the Krebs cycle. 

### 3.6. Prostatic Fluid Proteins and Zn

The human prostate glandular epithelium is known to secrete considerable amounts of proteins such as PSA, prostate secretory protein of 94 amino acids (PSP94) also known as β-microseminoprotein (not unique to the prostate), the unique human-specific transmembrane metalloprotease prostate-specific membrane antigen (PSMA) (also not unique to the prostate) and prostatic acid phosphatase [139,140]. The protease activity of PSA supports cancer cell invasion through the extracellular matrix [141]. Ishi et al., 2004 have shown the ability of Matrigel invasion by LNCaP prostate cancer cells [142]. The authors were also able to demonstrate the ability of an IC_50_ of 50 µM Zn^2+^ or 150 µM of Cd^2+^ to be able to suppress the specific activity of PSA purified from human plasma, while other divalent ions Cu^2+^, Mn^2+^, Fe^2+^, and Mg^2+^ were either unable to do such suppression or were weaker [142]. A 200 µg/d Se supplementation for six months in men without cancer from Auckland, New Zealand has shown that PSA suppression with supplementation was seen only in stratified groups of men including those having a Zn dietary intake above 14 mg/d [143].

The PSP94 protein is known for various aspects associated with prostate cancer suppression including growth inhibitory and apoptosis properties, suppressing tumor blood vessel density, inhibition of the matrix metalloproteinase secretion, and controlling bone metastasis [140]. Vanaja et al., 2003 report that the gene expression of both PSP94 and ZNF185 are inversely correlated with prostate cancer progression with the least amounts recorded in the metastatic disease followed by Gleason score 9, lymph node-positive disease; Gleason score 9, lymph node-negative disease; with the highest amounts recorded in adjacent benign tissues [129]. However, the direct impact of ZNF185 on PSP94 is not recorded yet. It is reviewed that binding the PSP94 protein to immunoglobulin IGg dampens the IGg-activated immunity required for functions of the male reproductive system [140]. Therefore, healthy rat prostate records only 0.3% of IGg compared to that of the serum [144]. However, local synthesis of IGg production in prostate cancer cells is higher compared to BPH, indicating a requirement to boost IGg immunoreactivity to combat cancer cells [145]. It is a possibility that it is for this reason that the PSP94 levels diminish with prostate cancer. The impacts of Zn deficiency and impaired ZIP10 transporter in IGg production are discussed in Hojyo and Fukada 2016 [146].

The PSMA protein has two Zn atoms at the active site coordinated by amino acid residues in the protease domain [139]. The review by O’Keefe et al., 2018 describes that PSMA expression is weak in the normal human prostate and increases with prostate cancer development [139]. The authors also have reviewed the PSMA action in recycling and uptake of folate in the growing prostate cancer cells; and its indirect involvement in protein kinase B (AKT) pathway activation and cancer aggressiveness as well as the correlation of PSMA levels with downstream effects of mammalian target of rapamycin (mTOR) pathway directed anabolic events [139]. Given the above pro-cancer functions of PSMA and the positioning of Zn in the active site of PSMA, it is a possibility that once prostate cancer sets in, an increase in the supply of Zn could also favor cancer progression.

### 3.7. Related Genetic Associations

In a Polish population, men with heterozygosity for the minor allele G of the metallothionine MT2A rs28366003 single nucleotide polymorphism (SNP) genotype are reported to have increased prostate cancer risk compared to the homozygous common allele carriers [147,148]. Gene expression level of MT2A was also lower among the minor allele carriers compared to average expression reported among homozygotes for the major allele, while prostate cancer tissue recorded higher levels compared to normal prostate tissue [147]. Levels of Zn, Cu, and Cd recorded from prostate cancer tissue too were higher among men carrying the minor allele G of the MT2A rs28366003 polymorphism. The author also reported an inverse correlation between prostate tissue Cu concentration and MT2A expression level. The allele G of the MT2A rs28366003 polymorphism showed an inverse association with blood measurements of Zn and Cu levels in a healthy group of men and women [149]. A systematic review of MMP genetic polymorphisms suggests that MMP3 11715A/6A and MMP9 rs17576 were associated with prostate cancer risk, while their meta-analysis shows no associations between MMP1 rs1799750, MMP2 rs243865, or MMP7 rs11568818 and overall prostate cancer risk [150].

According to Kodali et al., 2018 [151], rs1175550 SNP of the Small Integral Membrane Protein 1 (SMIM1)), rs2769264 of the ring finger protein 217 (RNF217), and rs2769270 (near Proteasome Subunit Beta 4 (PSMB4) and Selenium Binding Protein 1 (SELENBP1)) were associated with erythrocyte concentration of Cu. These authors further record that rs11638477 (Chromosome 15 Open Reading Frame 39 (C15orf39)), rs1532423 (Carbonic anhydrase 1 (CA1)), and rs2120019 (phosphopantothenoylcysteine decarboxylase (PPCDC)) were associated with erythrocyte concentration of Zn. A GWAS study by Ng et al., 2015 shows two chromosomal regions mapping to 4q24 and 1q41 associated with whole blood Mn levels [152]. According to these authors the SNP rs13107325 located in the 4q24 locus, which is in an exon of SLC39A8, which encodes ZIP-8 protein involved in manganese and zinc transport with the minor SNP associated with effects on protein formation, and the SNP rs1776029 located in the 1q41 locus in the intronic region of SLC30A10, coding for ZnT-10 protein associated with a possible Mn accumulation problem. These authors further report that loci 6q14.1 and 3q26.32 were associated with cadmium and mercury levels. 

Another GWAS study located an intergenic region near selenium binding protein 1 (SELENBP1; lead SNP rs17564336) and another in the intronic region of Ceruloplasmin gene (rs34951015) in association with blood Cu levels. The above indicates that genetic variation associated with Zn and other trace elements may add further complexity to the assessment of Zn requirements for maintaining a healthy prostate.

### 3.8. Zn Supplementation Effects

Based on the variation of tumor weights in the TRAMP model with variable dietary Zn supplementation, it has been shown that both deficiencies and excesses compared to optimum dietary requirements play a role in prostate carcinogenesis [153]. A deficiency of Zn in primary prostate epithelial cells has shown single-strand DNA breaks while altering the expression of genes responsible for DNA damage response [154]. In a cohort of Swedish men diagnosed with prostate cancer improved prostate cancer-specific survival was observed in the highest quartile of recorded dietary Zn intake of 15.6–20.1 mg/d compared to the lowest quartile of 9.0–12.8 mg/d [155]. According to a study of U.S. men participating in the Health Professionals Follow-Up Study taking supplemental Zn above 100 mg/d and men who took supplemental Zn for 10 or more years had an increased relative risk of advanced prostate cancer compared to non-users [156]. Continuous five-week supply of either a physiologic Zn supply with 1 µg/mL ZnSO_4_ or supraphysiologic Zn concentration of 10 µg/mL ZnSO_4_ in both LNCaP prostate cancer cells and PNT2 human normal prostate epithelial cells, showed increased expression of cancer-promoting genes in the latter [157]. The above indicates that unregulated Zn supplementation for prostate health benefits may create unwanted outcomes. A systematic review and meta-analysis show potential benefits of low-dose (≤25 mg/d) and long-duration (≥12 weeks) Zn supplementation on other health issues, each improving risk factors for type 2 diabetes and cardiovascular health than high-dose and short-duration interventions, respectively [158]. However, the bioavailable Zn reaching the prostate in the case of prostate cancer will be challenging [159]. Meanwhile, Singh et al., 2014 propose an evidence-based approach for resveratrol and Zn combination treatment for prostate cancer [160]. This group also discusses other dietary components such as quercetin, epigallocatechin-3-gallate, grape-seed procyanidin extract, and curcumin), that will enhance Zn uptake, accumulation, mobilization, regulation of ZnT efflux and ZIP influx transporters, and regulation of metallothionines as well as those that prevent Zn absorption (Phytate) [159]. 

A Mendelian Randomization study indicates that genetically predicted Zn was positively associated with ischemic heart disease [151]. Therefore, dietary or supplementary Zn adjustments that are considered beneficial for prostate health may have implications on cardiac health.

### 3.9. Interactions of Zn 

#### 3.9.1. Interaction with Other Elements

Gray et al., 2005 give a comprehensive account of the importance of Selenium (Se), Zn, and Cd in prostate cancer etiology [161]. Higher levels of blood Cd and Pb were recorded in men with elevated PSA compared to men with normal PSA before data was adjusted for confounding factors [162]. In a group of men taking part in NHANES 2001–2002, a positive association between serum PSA level and urinary Cd level was recorded only in men with lower dietary Zn intakes [163]. As mentioned before, men will have a reduction in PSA by a 200 µg/d Se supplementation for six months, only if their dietary intakes are above the recommended daily intake (RDI) for Zn [143]. Meanwhile, Cd-induced carcinogenesis due to its multiple impacts is known [164]. A review of four cancer types including prostate cancer shows that overall, cancer patients have elevated serum Cu and diminished Zn levels [165]. Alongside serum Zn levels, Se and Mn also showed decreases and were associated with increased Cu and Fe levels in men with prostate cancer compared to control groups [166]. A study with native African men from Nigeria records that healthy prostate tissue and serum of these men record higher levels of Zn^2+^ and lower levels of Cd^2+^, and lower Cd^2+^/Zn^2+^ ratios compared to men with prostate cancer [167]. Those with low dietary intakes of Fe, Zn, Ca, or protein show associations with dietary Cd uptake [168], which may indirectly affect Zn uptake, storage, distribution, and release from serum albumin and MT. Data assessed from the National Health and Nutrition Examination Survey (NHANES) 2007–2012 shows that those in the highest quintiles of Zn intake tended to record lower blood and urinary Cd concentrations [169]. Interaction with Fe and Ca reducing Zn absorption is also recorded [19]. 

The Nutritional Prevention of Cancer Study recorded beneficial effects of a 200 µg/d Se supplementation as selinized yeast given for an average of 4.5 y towards risk reduction of several cancers including prostate cancer in a group of men with a baseline plasma Se level of 114 ng/mL [170]. However, the subsequent Selenium and Vitamin E for cancer prevention study that used 200 µg/d Se supplementation as a more potent selenium source of selenomethionine could not reproduce the benefits in men recording a baseline serum Se level of 135 ng/mL [171]. The functions of Se towards cancer preventive properties are known [172] although beneficial levels may vary in association with demographic, lifestyle, genetic and dietary factors [143,173,174,175,176,177,178]. Unlike Zn which has an estimated RDA of 11.1 mg/d for human health, the estimated Se requirement for men ≥51 y is 55 µg/d [179]. Waters et al., 2018 have shown that both low and high levels of Se are unfavorable for health [180]. With the battery of seleno antioxidant enzymes required for human health, it is a possibility that Se benefits towards prostate cancer prevention take place with a minimal amount sufficient to optimize required enzymes to mop up the excess ROS produced in the prostate [143]. 

In a study by Tan and Chen 2011, a panel of nine trace elements (Zn, Cr, Mg, Ca, Al, P, Cd, Fe, and Mo) measured from hair samples was able to accurately classify (accuracy of 98.2%, a sensitivity of 100%, and a specificity of 96.4%) men with prostate cancer from healthy controls [181]. In another study by Guo et al., 2007, another panel (Zn, Mg P, K, Ca, Cr, Mn, Fe, Cu, and Se) measured from hair was able to predict prostate cancer with 95.8% accuracy [182]. Analyses of expressed prostatic fluid have shown that Zn and Rb are approximately 7.7 and 3.2 times, lower in the fluid of prostate cancer patients compared with those with BPH, while the investigated levels of Fe, Br, and Sr showed no difference [183]. Besides Zn, Se, Mn, Cu, and Fe are either at the activity center of several antioxidant enzymes or provide structural stability (e.g., seleno-enzymes including *GPX*s and thioredoxin reductases; superoxide dismutases; and catalases) and work in harmony to balance off oxidative stress in tissues while these trace elements also support many other functions including immunity [166]. In Cu-ZnSOD, Cu provides the catalytic activity while Zn provides the structural stability [19].

#### 3.9.2. Interaction of Zn with Demographic, Lifestyle, and Health Factors 

The requirement of trace elements for optimum health could vary with demographics and lifestyle [184], as well as the dietary and environmental exposure levels of heavy metals such as Cd, and therefore needs consideration when deciding on Zn requirements. In the NHANES 2007–2012 cohort, higher blood and urinary Cd were more pronounced in older subjects and those with lower BMI [169]. Reduced efficiency in the production of MT to sequestrate Cd with increasing age is suggested [75]. Serum Zn levels are inversely associated with the number of cigarettes smoked per day [185]. Serum, blood, and urinary levels of Cd are likely to be higher in smokers compared to non-smokers [169,186]. Heavy alcohol consumers and dietary supplement non-users record higher blood Cd levels [169]. Therefore, among such lifestyles, Zn may have interacting effects with Cd when competing for albumin-bound distribution, and storage in MT for subsequent use. Smokers also carry lower levels of serum Se [176]. Therefore, biological impacts of demographics and lifestyle on other trace element imbalances also could have indirect consequences in serum Zn levels, that may ultimately impact prostate health.

The Strong Heart Study and associated Strong Heart Family Study taken part by American Indian communities with baseline and two follow-up evaluations carried out approximately between 4 y and 8 y show that higher urinary Zn levels at baseline were associated with T2DM incidence [187]. The authors also record a similar association between urinary Zn level and pre-diabetes. It is a possibility that the prostatic Zn gets increasingly secreted out and eventually ends up in urine when levels of blood glucose or dextrose increase [188,189].

### 3.10. Transgender Women and Potential Impact on Zn Homeostasis

Surgical and medical interventions towards feminizing gender-affirming procedures in transgender women include bilateral orchiectomy, the use of a testosterone antagonist, GnRH agonist as well as simultaneous estrogen therapy [190]. However, transgender women retain their prostate gland even after gender-affirming surgery [191]. Therefore, maintenance of prostate health in transgender women is equally important as in cis men. A study of 10 case reports indicates that transgender women who have not opted for gender-affirming hormone therapy (GAHT) or who have not had gender-affirming surgery (GAS) have the same risk of prostate cancer as in the cis male population, while the risk of those opted for GAHT or GAS is lower than age-matched cis-males [192]. On the contrary, a study by Gaglani et al., 2022 suggests that transgender women on GAHT are at risk of aggressive prostate cancer than cis men [193]. The authors compare potential molecular pathways between cis men with prostate cancer treated with androgen deprivation therapies resulting in castration-resistant prostate cancer with transgender women treated with GAHT and GAS, ending up with the aggressive disease. 

In vitro studies with prostate cancer cell lines show associations between testosterone, ZIP-1, and Zn uptake, [51,72]. Furthermore, in prostate cancer cell line models, testosterone together with 1,25(OH)2D3, and 9 cis-retinoic acid are shown to redirect cytosolic citrate metabolism by restoring normal prostatic Zn homeostasis [194]. Thomas et al., 2014 transfected a PC-3 prostate cancer cell line (which is nuclear androgen receptor (AR)-negative with lower levels of ZIP-9) to overexpress ZIP-9 [195]. The authors demonstrated that treatment with testosterone upregulated proapoptotic genes and proteins in this cell line through the activation of a membrane AR. This could mean that treatment with testosterone-lowering drugs or surgery in transgender women might impair these proapoptotic functions if they carry ZIP-9 transporter-carrying prostate cancer cells. On the other hand, LNCaP prostate cancer cells that are inherited with both nuclear AR and ZIP-9, respond to testosterone by initiating metastatic migratory mechanism [66], and therefore testosterone lowering treatment in transgender women may be favorable in controlling metastasis if the cancer cells carry nuclear AR and ZIP-9. 

In a study with post-menopausal women receiving estrogen and progesterone conjugated therapy, Zn urinary excretion showed a declining trend only in those with an elevated Zn excretion at baseline, but not in those with normal urinary Zn excretion at baseline [196]. Whether such observations are comparable in transgender women treated with estrogen is not yet known. 

Zn is an important factor in bone health [197]. Although GAHT in adults has not shown major bone impacts, such treatments have recorded impaired bone development in adolescents [198]. However, whether such impairments occur through GAHT impacts on Zn uptake in transgender women is not yet known. Meanwhile, a review suggests a higher trabecular bone score and bone mineral density in the lumbar region in transgender women receiving GAHT [199].

It is important to improve understanding of Zn homeostasis in transgender women especially due to the potential impacts of their feminizing gender-affirming procedures on biological Zn components. So far, there is no recording of Zn levels in biological matrices in transgender women. 

### 3.11. Zn-Dependent Technologies Associated with Prostate Cancer

Using Zn assays on biopsy cores and the corresponding Gleason scores, Cortesi et al., 2010 have shown that tissue Zn depletion correlates with the Gleason score [200]. While the majority of Zn^2+^ is bound forms, there is a picomolar range existence of free Zn^2+^ found in the cytosol known for their intra- and intercellular communication [201]. Due to the importance of Zn in health, small molecule fluorescent Zn^2+^ probes are being tested to extend the evaluation of the cellular/subcellular Zn distribution and understanding of mechanisms taking place with dysfunctional Zn homeostasis in the prostate [11]. 

Costello and Franklin 2020 discuss the Zn ionophore Clioquinol treatment as a potential mechanism to deliver cytotoxic Zn levels to the ZIP-deficient malignant prostate cells and its success in controlling the androgen-independent malignant growth along with a prolactin inhibitory drug in an advanced prostate cancer patient [15].

Using magnetic resonance imaging with a gadolinium-based Zn-responsive contrast agent and glucose-stimulated Zn secretion has been detected in BPH tissue compared to prostate cancer in a dog model [189]. If this contrast can be adapted to be used on humans, that will support differentiating BPH from prostate cancer.

Recent studies with other cancers indicate that Zn-induced changes can be detected through the ratios of the stable isotopes of Zn (Zn^64^ and Zn^66^) in urine and plasma [202,203,204]. The theory behind these findings is that in the event of cancer, the heavier isotope Zn^66^ is retained in the tissue over the release of the lighter isotope. These may be useful for future prostate cancer evaluations as well.

A study by Ferro et al., 2021 has shown the value of the non-invasive, cheaper, easier to perform, FDA-approved, and CE-marked prostate health index (PHI) to predict positive biopsy in men even better than the multiparametric MRI (mpMRI) PI-RADS score, and comparable performance in the identification of clinically significant prostate cancer to the mpMRI procedure [205]. It will be of importance to evaluate correlations between the PSA level, PHI, mpMRI PI-RADS score of the prostate, and the disease severity with the levels of Zn in the prostate tissue as well as in biological matrices such as plasma, serum, seminal plasma, hair, and urine to understand whether Zn levels can enhance current prostate cancer predictions.

## 4. Discussion and Commentary

This review was made to collate multiple features of the involvement of Zn in prostate health and disease to understand the applicability of the knowledge toward aspects of prostate cancer-related events. The major requirement of Zn for prostate health is its ability to truncate the Krebs cycle to prevent citrate oxidation and associated functions [47,48]. However, Zn is also involved in multiple other functions including Zn signaling towards regulating prostate cancer outcomes [5] accompanied by multiple Zn influx and efflux transporters [57], storage in MTs [102,103], Zn finger involvement for protein regulation functions [119,120], and Zn coordinated MMP activity that facilitates cancer cell invasion [117]. A summary of the potential points between Zn ingestion to excretion which may impact prostate cancer outcomes reviewed is given in Figure 2. Even if a man takes the RDA for Zn, whether a sufficient level will reach for optimized prostate functions and health will also depend on a myriad of other factors including other trace element involvement, demographic, lifestyle, and health status.

This review indicates the complexity of finding a dietary/supplemental Zn level or a circulatory Zn level for optimizing prostate health. The factors are further complicated by the presence of genetic variability associated with multiple events discussed in this review (e.g., [147,148]). Both deficiencies and excesses of Zn nutrition may have implications on prostate cancer outcomes. However, advanced data modeling with multiple relevant factors may potentially provide a solution to decide on a more personalized Zn requirement for better prostate health in the future. Until such time it is important to understand how to optimize the use of ingested Zn with lifestyle modifications and good health. These include cessation of tobacco smoking habits that cause higher systemic Cd burden [169,186] and compete with Zn binding sites on albumin and MTs; changing excessive alcohol consumption lifestyle known to associate with higher blood Cd levels [169]; refraining from excessive intakes of high-fat diets to prevent competition on Zn circulation through albumin [32] and preserve albumin structure for proper function [33]; minimizing stress to reduce ROS to facilitate stability of ZF proteins [124]; increasing consumption of fermented or germinated food products to avoid phytates to support Zn binding to albumin for circulation [20]; manage blood sugar levels to minimize diabetes or pre-diabetes which are known to associate with Zn urinary excretion [187]; increased plant-based diets to improve Zn transporter regulation by the phytochemical action [159], and plant fiber to increase gut transit time to optimize Zn absorption.

Although this review cannot support a ‘one size fits all’ solution for optimal Zn intakes for prostate health, the evidence may be useful in multiple directions for prostate cancer prevention, diagnosis, prognoses, and treatment events alongside standard clinical practices. Level of serum Zn may be useful alongside other factors such as genetics, demographic and lifestyle, comorbidities, family history of cancer, previous biopsy details, and interacting trace element profiles for developing algorithms to improve the efficiency of current prostate cancer screening marker PSA or more effective methods such as PHI, and mpMRI PI-RADS score [54,205,208,209]. Post-radical prostatectomy prostate tissue analyses for levels of Zn will improve understanding of the extent of malignancy and prognosis based on tissue Zn level and distribution alongside standard clinical grading and staging of the disease [54]. Prostate tissue Zn level and distribution assessed through advanced imaging technologies in dog models were able to differentiate between prostate cancer and BPH [189]. If this is translated to be implemented in men, may help substantiate the differentiation between prostate cancer and BPH. Additionally, Zn transporter protein distribution [53,67,90,91], and matrix metalloproteinase activity assessed through histochemistry of resected tissue [117] may also support understanding the disease extent which will further support prostate cancer management decisions. Body Zn distribution is highest in muscle (57%) followed by bone (29%) [210]. Both these tissues are important in maintaining the healthy physical activity of aging men, especially in men treated with ADT for advanced prostate cancer [97,211,212]. O’Connor et al., 2020 give an account of the role of Zn in bone growth, homeostasis, and regeneration [197]. ZnT-5 deficiency-related bone impacts are seen in mice [96], and if this can be proven in humans, it may be useful to stratify men at diagnoses for prostate cancer management protocols towards minimizing bone-impacting ADT therapies [98,213,214] which will be needed to manage advance prostate cancer. 

The Zn efflux transporter ZnT8 is known for the specific transport of Zn to insulin secretory granules of the pancreatic β cells for the required release of insulin [215]. Galvez-Fernandez et al., 2022 provide information on the involvement of Zn association with multiple factors leading to type II diabetes including the influence of a variant of the *SLC30A8* gene encoding this protein [187]. In a study by these authors, men with type II diabetes or pre-diabetes showed increased Zn urinary excretion [187]. Therefore, at the time of prostate cancer diagnoses, men with type II diabetes or pre-diabetes may require additional clinical support to adequately control blood sugar levels, to suppress diabetes-associated Zn urinary excretion which may otherwise further assist in prostate cancer progression. Meanwhile, it is considered that aspects of ZF protein signaling domains, may carry the potential to be developed as prognostic markers as well as therapeutic targets in prostate cancer [54,130].

According to Costello and Franklin 2006, Zn content in a prostatic fluid is 8000–10,000 nmol/g [14]. Prostatic fluid is estimated to contribute 37–44% of the seminal plasma [216]. Therefore, every shedding of semen could be removing a considerable amount of prostatic Zn. However, in the Health Professionals Follow-up Study of 31,925 men, with an 18 y of follow-up and utilizing questionnaire-based self-reported data records that ≥21 ejaculations/month during ages 20–29 y and 40–49 y reduced prostate cancer risk especially that of low-risk disease compared to 4–7 ejaculations/month [217]. However, based on data associated with the questionnaire completion in the year before prostate cancer diagnoses, there was some evidence that ejaculation frequencies ≥21/month were associated with advanced or lethal prostate cancer. 

It may be beneficial to understand the levels of elements in biological matrices that will interact with Zn and suppress the Zn supply needed for prostate health including Cd and those elements which are components of a battery of antioxidant enzymes (Se, Cu, Fe, Mn) that have to act in harmony towards free radical removal [161,162,163,166,181,182]. 

Developing imaging methods may be the future non-invasive means of understanding Zn tissue distribution in the prostate [11,189] that will further support clinical decisions beyond PSA-based prostate cancer screening.

Some aspects of Zn’s involvement in prostate health and disease are not yet known. For example, although IMA levels are implicated in many cancers including prostate cancer [39], the interaction of serum IMA and Zn levels is not yet known. Similarly, although higher serum levels of free fatty acids are associated with prostate cancer, prostate disease outcomes through their impact on the Zn carrier capability of albumin are not known. Although studies indicate a general reduction of ZIP proteins with prostate cancer malignancy, a correlation between ZIP protein expression levels and prostate cancer grades is not yet recorded [70]. Studies with breast cancer cell lines indicate the function of ZIP10 and ZIP6 in EMT [71]. Various aspects of breast cancer progression association with Zn influx transporters have also been reviewed [70]. However, such information is not available for prostate cancer. ZnT-2 expression correlation with Zn level in the dorsolateral lobe of the prostate is reported from a rodent model [94], but not yet with prostate cancer. 

## 5. Conclusions

The influence of Zn on prostatic health is multidimensional. Due to various factors influencing Zn bioavailability and transport to the prostate, algorithms toward more personalized Zn requirements have not yet been achieved other than the Zn RDA. Optimizing the ingested Zn for prostate and general health can be achieved with lifestyle modifications and health. However, there are multiple opportunities to utilize and improve the understanding of Zn associations with prostate health and disease. 

## Figures and Tables

**Figure 1 biomedicines-10-03206-f001:**
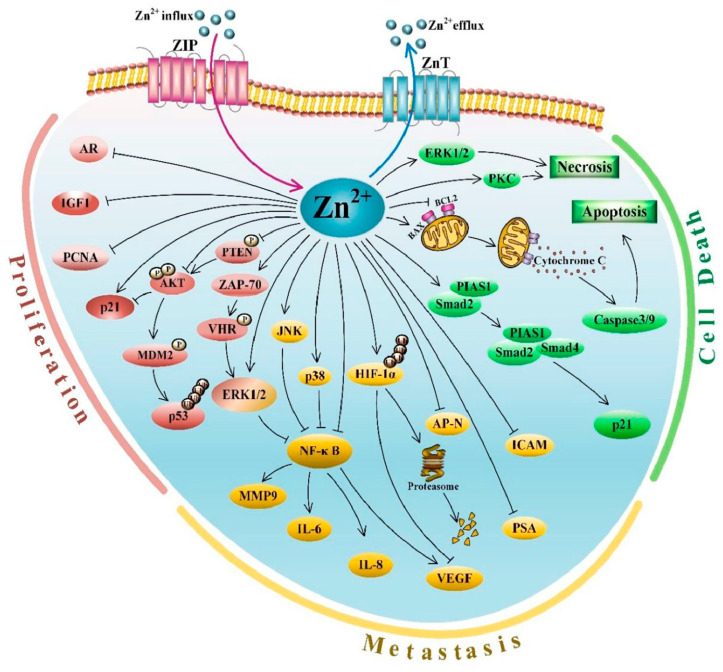
An overview of molecular signaling pathways mediated by Zn in prostate cancer. Zn is involved in various antiproliferative and proapoptotic pathways to exert its antitumor activities, including suppressing cell proliferation, inducing cell death, and inhibiting metastasis. Zn-regulated molecules involved in proliferation, metastasis, and cell death, as well as Zn transporters, are grouped and differentially colored. Black T bar: inhibiting the function of Zn or proteins; Blue arrow: transporting Zn from cytoplasm to extracellular fluid; Purple arrow: transporting Zn from the extracellular fluid into the cytoplasm; Black arrow: activating function of Zn or proteins (Adapted from Li et al., 2020 [54] under the Creative Commons License 4.0 International (CC BY 4.0)).

**Figure 2 biomedicines-10-03206-f002:**
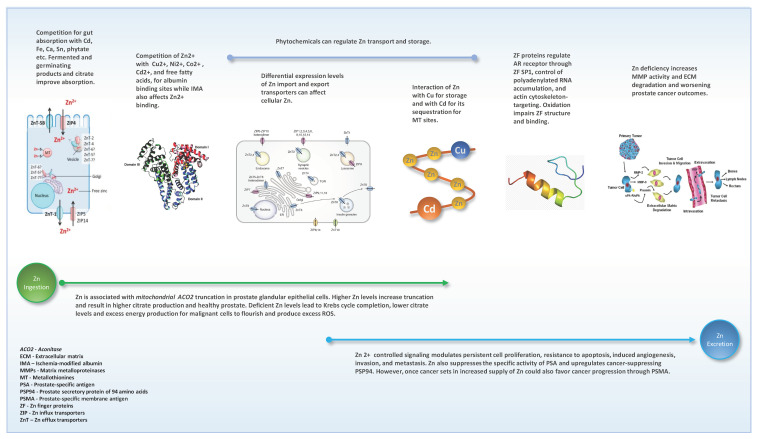
Potential points between Zn ingestion to excretion where Zn absorption, distribution, influx, efflux, and storage may be affected and subsequent impacts on Zn signaling and prostate cancer outcomes. (Images were adapted from Maares and Haase 2022 [20] for Zn gut absorption (under the Creative Commons License 4.0 International (CC BY 4.0), Stewart et al., 2003 [22] for Zn albumin binding (with Copyright (2003) from National Academy of Sciences, U.S.A), Kambe et al., 2021 [57] for Zn transporters under the Creative Commons License 4.0 International (CC BY 4.0), from [206] for ZF SP1 under the Creative Commons Attribution License 3.0 (CC BY 3.0), and Dasgupta et al., 2012 [207] for MMPs-related prostate cancer outcomes under the Creative Commons Attribution License 3.0 (CC BY 3.0).

## Data Availability

Not applicable.

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
