# Peer review of "Zinc in Prostate Health and Disease: A Mini Review"

_biomedicines, 2022, doi:10.3390/biomedicines10123206_

Round 1
Reviewer 1 Report
The article talks about the risk factors associated to prostate cancer: from diet to lifestyle, focusing mainly on trace elements, in particular zinc (Zn), and on the role they play in promoting or preventing prostate disease. The recommended daily amount of zinc is 11mg, but there are multiple factors that influence its absorption, secretion and excretion to determine it certainty. Zinc has multiple roles including forming transport proteins, inflow and outflow channels, metallothionins, the metalloprotease matrix, Zn finger proteins and an intracellular signaling role. For example, ZIP-1 and ZIP-4 zinc intracellular transporters are less expressed in tumor tissue than in benign prostatic hypertrophy (BPH). There is also an increase in matrix metalloproteinases (MMPs) responsible for the degradation of cells of the extracellular matrix to promote metastasis. Lower amounts of proteins such as PSP94 and ZNF185 are also present in prostate fluid in patients with prostate cancer. Moreover nteractions of zinc with other ions such as copper (Cu), iron (Fe) and manganese (Mn) are analyzed. In conclusion, the correct dose of zinc (Zn) to take daily is not clear yet, given the great inter-individual variety, and it is not possible to use zinc and all the zinc-containing proteins seen above as a tumor marker without carrying out further studies.
My suggestions:
- English language should be improved in both grammar and syntax.
- Section "3.2 Zn absorption from gut and blood circulation"
Exposes about increased serum levels of IMA (ischemia-modified albumin) associated with lower levels of zinc (Zn) in patients with prostate cancer gleason> 7; but the levels also rise in the patient with psoriasis and hypertension as well as in other cancers.
It is recommended to removed from the specific review this section and elaborate it separately in another study.
-Section 3.8 "Zn supplementation effects"
talks about the role of zinc (Zn) and copper (Cu) in ischemic heart disease (IHD)
I recommend to delete this part because it is of little interest to the article.
-Section 3.10 “Zn dependent technologies associated with prostate cancer”
Talks about the zinc (Zn) levels in relation to the Gleason score:
It is possible to perform an analysis to correlate the prostate health index and multiparametric MRI of the prostate (with its PIRADS scale) to the levels of zinc in the tissues and to the severity of the disease.
At this regard i can suggest this work:
Https://pubmed.ncbi.nlm.nih.gov/34572950/
-Section "Discussion and commentary"
The amount of zinc (Zn) detected in post prostatectomy tissues correlates with the degree of disease extension, but the cited article “185: Khalighinejad P, Parrott D, Jordan VC, Chirayil S, Preihs C, Rofsky NM, et al. Magnetic Resonance Imaging Detection of Glucose-Stimulated Zinc Secretion in the Enlarged Dog Prostate as a Potential Method for Differentiating Prostate Cancer From Benign Prostatic Hyperplasia. Invest Radiol. 2021; 56 (7): 450-7 ” talks about on studies carried out on animals.
Have similar studies been carried out in humans as well? Otherwise I suggest this type of study.
Author Response
Please note my comments to Reviewer 1 are attached.

Reviewer 2 Report
In this paper the knowledge on the effects of zinc on prostate health and prostate disease is reviewed.
General comment: The topic of the review is relevant and the review is comprehensive and well structured.
Specific points:
Title: 'Zinc and prostate health and prostate disease: a mini review'. It could be 'Zinc and prostate health and disease: a mini review'.
Figure 1: It is written 'prolifiration' instead of 'proliferation'.
Figure 1 is not reffered to in the main text.
Figure 2: Its size should be increased, or the font size should be increased.
Reviewer 3 Report
The review is very extensive and the author has done a great job of writing the review on zinc and prostate cancer. It includes the role of Zn in healthy prostate, prostate cancer, prevention, as well as interaction with other elements.
Comments:
1. Please make the font size larger in Fig.2 it is hard to read the text in the figure.
2. The review does a great job of compiling information on correlation of zinc and prostate health, along with prostate cancer in men. Information on transgender women is missing, and it would be great to make the review more gender inclusive.
3. The author could add effect of Zn on ROS, DNA integrity, etc. in prostate cancer in the Zinc signaling section. Reference to Fig 1 is missing in the text. A slight expansion of the figure 1 legend in the text would be helpful.
Round 2
Reviewer 1 Report
Authors answered all comments and suggestions.